# NHR-8 and P-glycoproteins uncouple xenobiotic resistance from longevity in chemosensory *C. elegans* mutants

Gabriel A Guerrero[1†], Maxime J Derisbourg[1*†], Felix AMC Mayr[1], Laura E Wester[1], Marco Giorda[1], J Eike Dinort[1], Matías D Hartman[1], Klara Schilling[1], María José Alonso-De Gennaro[1], Ryan J Lu[2], Bérénice A Benayoun[2], Martin S Denzel[1,3,4*]

[1]Max Planck Institute for Biology of Ageing, Cologne, Germany; [2]Leonard Davis School of Gerontology, University of Southern California, Los Angeles, Los Angeles, United States; [3]CECAD - Cluster of Excellence University of Cologne, Cologne, Germany; [4]Center for Molecular Medicine Cologne (CMMC), University of Cologne, Cologne, Germany

**Abstract** Longevity is often associated with stress resistance, but whether they are causally linked is incompletely understood. Here we investigate chemosensory-defective *Caenorhabditis elegans* mutants that are long-lived and stress resistant. We find that mutants in the intraflagellar transport protein gene *osm-3* were significantly protected from tunicamycin-induced ER stress. While *osm-3* lifespan extension is dependent on the key longevity factor DAF-16/FOXO, tunicamycin resistance was not. *osm-3* mutants are protected from bacterial pathogens, which is *pmk-1* p38 MAP kinase dependent, while TM resistance was *pmk-1* independent. Expression of P-glycoprotein (PGP) xenobiotic detoxification genes was elevated in *osm-3* mutants and their knockdown or inhibition with verapamil suppressed tunicamycin resistance. The nuclear hormone receptor *nhr-8* was necessary to regulate a subset of PGPs. We thus identify a cell-nonautonomous regulation of xenobiotic detoxification and show that separate pathways are engaged to mediate longevity, pathogen resistance, and xenobiotic detoxification in *osm-3* mutants.

*For correspondence:
mderisbourg@age.mpg.de
(MJD);
mdenzel@age.mpg.de (MSD)

†These authors contributed
equally to this work

Reviewing Editor: Elçin Ünal,
University of California, Berkeley,
United States

## Introduction

Chemosensation is a genetically tractable phenotype in various organisms. In *Caenorhabditis elegans*, many mutants with defective chemosensation have been identified. Sensory phenotypes are complex in nature, and many of the classical chemosensory mutants were originally characterized by their behavioral phenotypes. Odr mutants, for example, have an abnormal odorant response, while Osm (osmotic avoidance abnormal) mutants do not avoid high-salt environments (*Bargmann, 2006*). Mutations in components of neuronal G-protein coupled receptor (GPCR) signaling, such as *tax-4* and *odr-1*, cause atypical chemosensory behavior (*Bargmann, 2006*). These mutants are not only characterized by a failure to adequately respond to their environment, but show additional phenotypes linked to various life traits including pathogen resistance, increased lifespan, and drug detoxification (*Gaglia et al., 2012*; *Apfeld and Kenyon, 1999*; *Dent et al., 2000*). Many chemosensory mutants are long-lived, and this phenotype depends on the DAF-16/FOXO transcription factor that is regulated by the insulin signaling pathway (*Apfeld and Kenyon, 1999*).

Beyond mutations in GPCRs, variants in genes involved in the development of amphid sensory neurons can lead to chemosensory defects. One example are mutations in intraflagellar transport (IFT) proteins that prevent full cilia development of amphid sensory neurons (*Inglis et al., 2007*).

Alterations in the IFT genes *osm-3* and *daf-10* disrupt the development of neuronal dendrites that project toward the tip of the nose, where they are exposed to the outer environment (*Inglis et al., 2007*). Thus, IFT mutants are usually characterized by defects in the chemical perception of their environment. Furthermore, developmental defects in lumen formation of the amphid head channel in *daf-6* mutants prevent the direct contact of amphid sensory neurons with the outside (*Perens and Shaham, 2005*), resulting in chemosensory defects.

In wild-type (WT) worms, amphid and phasmid sensory neurons can be visualized using the lipophilic dye DiI (1,1'-dioctadecyl-3,3,3',3'-tetramethylindocarbocyanine perchlorate), which fluorescently stains them bright red. The dye is passively taken up by sensory neurons when they are fully developed and in contact with the environment. Dye filling defective (Dyf) *C. elegans* are defined by an inability to take up DiI into their sensory neurons (*Inglis et al., 2007*). The Dyf phenotype arises in *daf-6* animals and also in many IFT mutants. Interestingly, Dyf mutants have unique stress resistance phenotypes that act via different signaling pathways. One such Dyf mutant, *daf-10(m79)*, has a unique resistance to pathogenic bacteria that was proposed to be downstream of sensory input (*Gaglia et al., 2012*). Furthermore, pairs of amphid head neurons, such as ASH and ASJ that often become disrupted in Dyf mutants, help coordinate the innate immune response to bacterial stress after infection (*Meisel et al., 2014*). Longevity is often associated with activation of physiological stress response pathways and has been shown to be regulated by the insulin signaling pathway in Dyf *C. elegans* (*Apfeld and Kenyon, 1999*).

Conserved mechanisms of longevity and stress resistance have additionally been linked to hyposmia in higher model organisms. In *Drosophila melanogaster*, loss of the putative chemoreceptor *Or83b* has been shown to significantly increase lifespan (*Libert et al., 2007*). Moreover, mice with ablated olfactory sensory neurons have a unique metabolic signature that protects them from deleterious effects of a high-fat diet (*Riera et al., 2017*).

While the olfactory machinery is seemingly more complex in higher organisms, *C. elegans* have unique phenotypes associated with the loss or alteration of their sensory neurons. First, unbiased screens for drug resistance have independently identified the Dyf phenotype as a common feature in drug-resistant *C. elegans* (*Fujii et al., 2004*; *Ménez et al., 2016*; *Collins et al., 2008*). Selection for resistance to the anthelmintic drug ivermectin was shown to enrich for Dyf mutants in several forward genetic screens (*Ménez et al., 2016*; *Page, 2018*; *Dent et al., 2000*). Follow-up work identified P-glycoproteins (PGPs), a class of ATP binding cassette (ABC) transporters, as the drivers behind ivermectin resistance in Dyf mutants (*Ardelli and Prichard, 2013*). Second, some chemosensory mutations, such as in *daf-10*, lead to resistance to pathogenic bacteria (*Gaglia et al., 2012*).

Here, we characterize a long-lived Dyf mutant, *osm-3*, that was previously identified in a forward genetic screen for resistance to the potent ER stressor tunicamycin (TM). From this screen, we reported the role of the hexosamine biosynthetic pathway in TM resistance and longevity (*Denzel et al., 2014*). Our aim was to uncover the mechanism by which Dyf mutants are resistant to TM. We found that the TM resistance of *osm-3* mutants is independent from its longevity and pathogen resistance phenotypes. Furthermore, we showed that PGPs mediate TM resistance in Dyf animals, and we found that the nuclear hormone receptor *nhr-8* mediates *osm-3* drug resistance. The 'Green Theory' of aging proposes that the accumulation of toxic endo- and xenobiotic debris contributes to physiological decline with age (*Afschar et al., 2016*; *Gems and McElwee, 2005*). Our work highlights drug resistance as a unique characteristic of Dyf mutants and functionally uncouples it from known stress response pathways that have been implicated in stress resistance and longevity of chemosensory mutants.

## Results

### Dye filling defective *C. elegans* mutants are resistant to TM

Tunicamycin (TM) is commonly used to induce endoplasmic reticulum (ER) stress by inhibiting the addition of N-glycans to nascent polypeptides (*Parodi, 2000*; *Heifetz et al., 1979*). In *C. elegans*, where the ER machinery is conserved, treatment with TM is toxic and induces the ER unfolded protein response. In fact, on plates containing TM at concentrations above 4 μg/mL, newly hatched larvae die at early developmental stages (*Shen et al., 2001*). Previously, we had carried out an ethyl methanesulfonate mutagenesis screen to identify long-lived mutants using TM resistance in *C. elegans* as

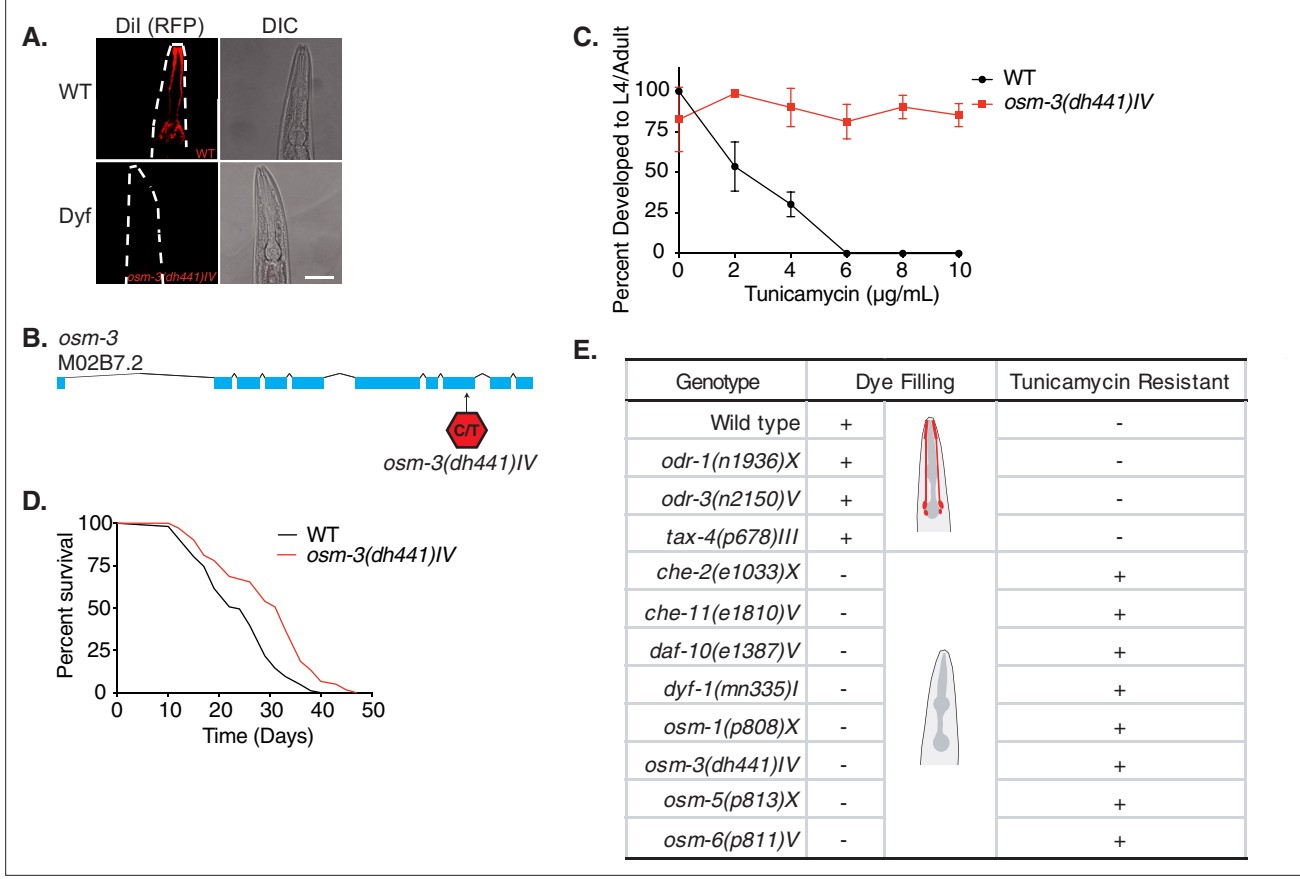

**Figure 1.** Tunicamycin resistance occurs in dye filling defective long-lived chemosensory *C. elegans* mutants. (**A**) Fluorescence and differential interference contract confocal microscopy images of WT and *osm-3(dh441)IV C. elegans* after DiI treatment (scale bar 40 μm). (**B**) Depiction of the *osm-3(dh441)* locus. *osm-3(dh441)IV* has a C to T mutation at position 3796925 of exon 8. (**C**) Developmental tunicamycin (TM) resistance assay using the indicated concentrations with WT and *osm-3(dh441)IV* animals. Data are mean ± SD (n = 3 independent experiments with >15 scored animals each). (**D**) Demographic lifespan analysis of WT and *osm-3(dh441)IV* animals. WT mean lifespan = 24 days, *osm-3(dh441)IV* mean lifespan = 29 days, p<0.0001. See *Supplementary file 1* for statistical analysis. (**E**) Table of dye filling phenotype and TM resistance. In the dye filling column,+ is positive for DiI filling as in (**A**) and – is Dyf. In the TM resistant column, + is resistant and – is not resistant to 10 μg/mL TM.

The online version of this article includes the following figure supplement(s) for figure 1:

**Source data 1.** Developmental TM resistance assay with WT and *osm-3(dh441)* animals (*Figure 1C*).

**Figure supplement 1.** Tunicamycin resistance occurs in dye filling defective long-lived chemosensory *C. elegans* mutants.

**Figure supplement 1—source data 1.** Developmental TM resistance assay with WT and *che-2(e1033)X* animals.

**Figure supplement 1—source data 2.** Developmental TM resistance assay with WT and *che-11(e1810)V* animals.

**Figure supplement 1—source data 3.** Developmental TM resistance assay with WT and *daf-10(e1387)V* animals.

**Figure supplement 1—source data 4.** Developmental TM resistance assay with WT and *dyf-1(mn335)I* animals.

**Figure supplement 1—source data 5.** Developmental TM resistance assay with WT and *osm-1(p808)X* animals.

**Figure supplement 1—source data 6.** Developmental TM resistance assay with WT and *osm-5(p813)X* animals.

**Figure supplement 1—source data 7.** Developmental TM resistance assay with WT and *osm-6(p811)V* animals.

**Figure supplement 1—source data 8.** Developmental TM resistance assay with WT and *odr-1(n1936)X* animals.

**Figure supplement 1—source data 9.** Developmental TM resistance assay with WT and *odr-3(n2150)V* animals.

**Figure supplement 1—source data 10.** Developmental TM resistance assay with WT and *tax-4(p7689)III* animals.

a proxy phenotype for longevity (*Denzel et al., 2014*). The largest cohort of classifiable mutants presented the Dyf phenotype (*Figure 1A*).

Among the Dyf mutants from the TM resistance screen, we found new alleles of genes previously linked to ciliary development. The mutant allele we selected for our investigation, *osm-3(dh441)IV*

(this allele will be referred to hereafter as *osm-3)*, carries a premature stop in the eighth exon, which prevents normal cilia development causing the Dyf phenotype (*Figure 1A,B*). *osm-3* is a member of the kinesin family involved in axonal development and transport, and mutants are known to show a Dyf phenotype (*Inglis et al., 2007*). While WT worms fail to develop on TM at concentrations above 4 μg/ mL, *osm-3* mutants fully develop at concentrations at least up to 10 μg/mL TM (*Figure 1C*; *Figure 1— source data 1*). Loss of *osm-3,* as well as other Dyf mutations, promote longevity in *C. elegans* (*Apfeld and Kenyon, 1999*). Demographic lifespan analysis of *osm-3* indeed confirmed extended lifespan in our point mutant (*Figure 1D* and *Supplementary file 1*). *osm-3* mutants, and Dyf mutants in general, have never been described as TM resistant especially at concentrations as high as 10 μg/mL.

To further explore whether the Dyf phenotype is linked to TM resistance, we tested other Dyf mutants that were previously described as long-lived (*Apfeld and Kenyon, 1999*). To our surprise, all long-lived Dyf mutants that we tested proved to be significantly TM resistant at the WT lethal dose of 10 μg/mL TM (*Figure 1E*, *Figure 1—figure supplement 1A–G*, *Figure 1—figure supplement 1—source data 1–7*). In contrast, the long-lived chemosensory *odr-1(n1936)X, odr-3(n2150)V,* and *tax-4(p678)III* mutants that do not display a Dyf phenotype were only slightly TM resistant (*Figure 1E*, *Figure 1—figure supplement 1H–J*, *Figure 1—figure supplement 1—source data 8–10*). Our data thus suggest that, among the larger class of chemosensory defective mutants, lifespan extension is uncoupled from TM resistance.

## Tunicamycin-Induced ER stress response is blunted in *Osm-3* mutants

TM treatment activates the ER unfolded protein response (UPR[ER]); we thus characterized the overall transcriptional response of *osm-3* mutants upon TM treatment. We performed transcriptomic analyses of *osm-3* and WT worms after 6 hr of TM treatment. Notably, the gene ontology (GO) terms related to ER stress and ER protein folding showed significant upregulation in WT animals, but not in *osm-3* mutants (*Figure 2A*). RT-qPCR analysis of UPR target genes further confirmed that there was no UPR[ER] induction in TM-treated *osm-3* mutants compared to the WT (*Figure 2—figure supplement 1*, *Figure 2—figure supplement 1—source data 1*). A hallmark of UPR[ER] activation in *C. elegans* is the upregulation of the molecular chaperone HSP-4/BiP (*Shen et al., 2001*). Upon TM treatment of *osm-3* mutants carrying an *hsp-4*::GFP reporter construct, there was no significant increase in the GFP signal compared to the untreated control, while in the WT, the GFP levels were significantly increased (*Figure 2B,C*, *Figure 2—source data 1*). This observation corroborates the results from the transcriptome analysis.

One potential explanation for a suppressed UPR[ER] response in *osm-3* mutants might be a general defect in the stress signaling pathway. To address this possibility, we sought to activate the UPR by alternate means. The ER membrane protein SEL-1/HRD3 is a cofactor of the HRD1 ubiquitin ligase complex involved in ER-associated degradation (ERAD) (*Hampton et al., 1996*). Using *sel-1* RNAi to inhibit ERAD, we observed a robust activation of the *hsp-4*::GFP reporter, suggesting a functional UPR[ER] response upon ER stress in *osm-3* mutants (*Figure 2D,E*, *Figure 2—source data 2*). We further used DTT treatment to induce ER stress and found normal levels of *hsp-4*::GFP induction and *xbp-1* splicing (*Figure 2F*, *Figure 2—source data 3*) in *osm-3* mutants. Together, these observations rule out that the reduced response to TM observed in *osm-3* mutants was due to a defect in UPR signaling.

## TM resistance in *Osm-3* mutants is independent from *Daf-16* or the PMK-1/p38 MAPK pathway

The lifespan extension observed in chemosensory-defective *C. elegans* as well as in *Drosophila* has been shown to be at least partially insulin signaling dependent (*Apfeld and Kenyon, 1999*; *Libert et al., 2007*). Therefore, we performed a demographic lifespan analysis to determine the role of the insulin signaling pathway in the lifespan extension of *osm-3* mutants. Indeed, the *osm-3* lifespan extension was fully *daf-16* dependent, as the lifespan of the *osm-3; daf-16* double mutants was identical to the *daf-16* lifespan (*Figure 3A*, *Supplementary file 1*). In stark difference to *osm-3* survival, the *osm-3; daf-16* double mutant remained resistant to TM while *daf-16* single mutants do not develop at 10 μg/mL TM (*Figure 3B*, *Figure 3—source data 1*). A previous study by *Henis-Korenblit et al., 2010* had shown elevated *hsp-4*::GFP levels during *daf-16* knockdown in a *daf-2* mutant. We therefore asked whether the knockdown of *daf-16* would sensitize *osm-3* mutants to TM. Interestingly, knockdown of *daf-16* by RNAi did not elevate the *hsp-4*::GFP response of *osm-3* mutants after TM

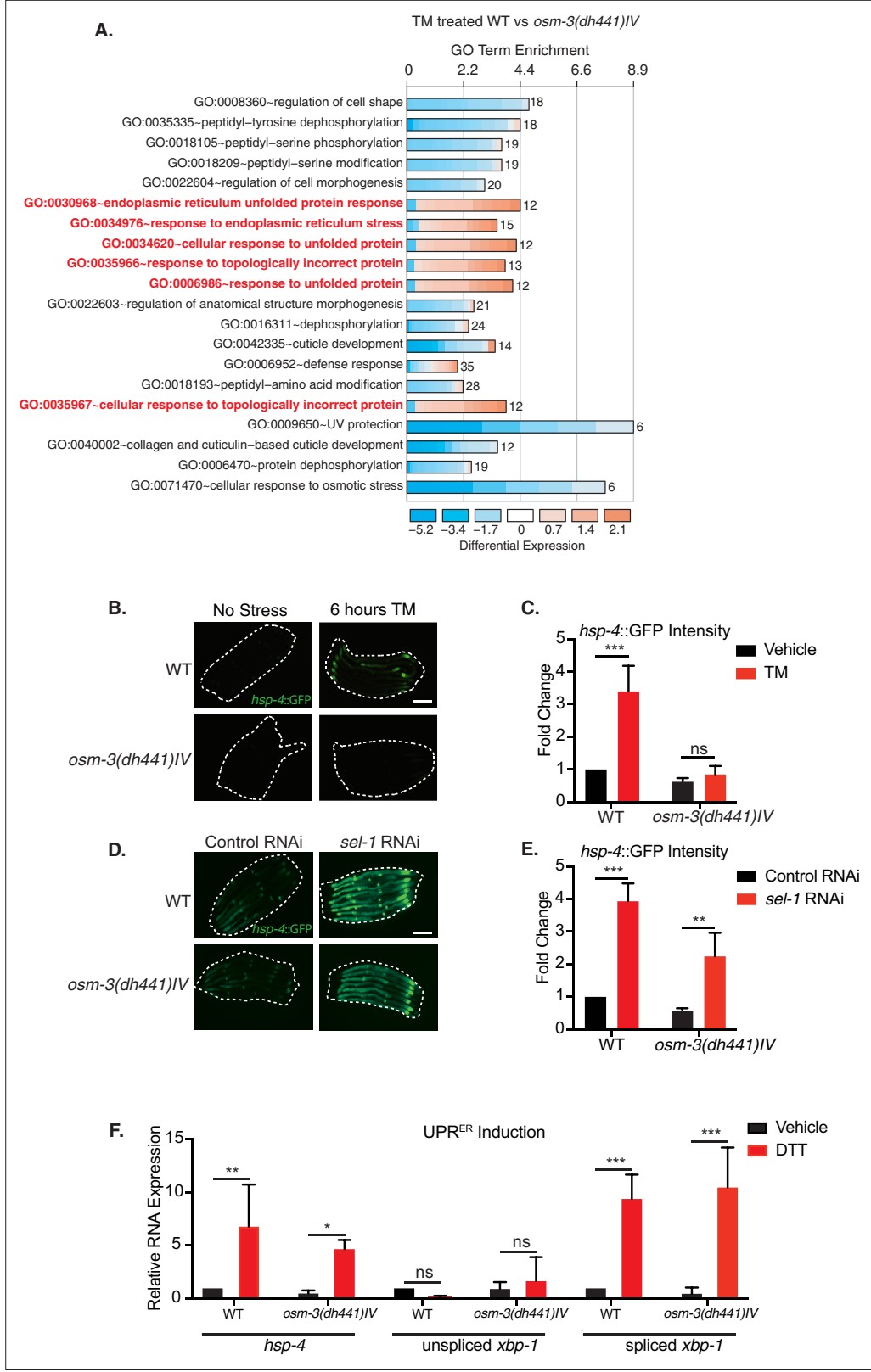

**Figure 2.** Reduced tunicamycin-induced ER stress signaling in *osm-3* mutants despite functional ER-UPR pathway. (**A**) DAVID gene ontology (GO) terms that are enriched in TM-treated WT compared to TM-treated *osm-3(dh441) IV* worms. Red = upregulated and blue = downregulated. The fold enrichment is plotted as x axis. The numbers of genes is indicated next to the bar for each term. (**B**) Green fluorescent images of WT and *osm-3(dh441)IV* animals

*Figure 2 continued on next page*

*Figure 2 continued*

in the *hsp-4*::GFP reporter background after 6 hr of 10 µg/mL TM treatment. Worms are outlined in the images (scale bar 100 µm). (**C**) Biosorter analysis of *osm-3(dh441)IV* vs. WT animals in the *hsp-4*::GFP background after 6 hr of control or TM treatment. Fold change represents GFP intensity normalized to WT treated with vehicle. Data are mean + SEM, n = 4, ***p<0.0001 by two-way ANOVA. (**D**) Green fluorescent images of WT and *osm-3(dh441) IV* animals in the *hsp-4*::GFP background after development on control or *sel-1* RNAi. Worms are outlined in the images (scale bar 100 µm). (**E**) Biosorter analysis of *osm-3(dh441)IV* vs. WT animals in the *hsp-4*::GFP background raised on control or *sel-1* RNAi. Fold change represents GFP intensity normalized to WT treated with vehicle. Data are mean + SEM, n = 3, **p<0.005 ***p<0.0001 by two-way ANOVA. (**F**) Quantitative PCR measuring relative *hsp-4* and spliced and unspliced *xbp-1* mRNA levels in WT and *osm-3(dh441)IV animals* after 2 hr of 10 mM DTT treatment. Relative mRNA expression is mRNA expression levels normalized to WT treated with vehicle. Data are mean + SEM, n = 3, *p<0.05, **p<0.001, ***p<0.0001 by two-way ANOVA.

The online version of this article includes the following figure supplement(s) for figure 2:

**Source data 1.** Biosorter analysis of *osm-3(dh441)IV* vs. WT animals in the hsp-4::GFP background after 6 hr of control or TM treatment (*Figure 2C*).

**Source data 2.** Biosorter analysis of *osm-3(dh441)IV* vs. WT animals in the hsp-4::GFP background raised on control or *sel-1* RNAi.

**Source data 3.** Quantitative PCR measuring relative *hsp-4* and spliced and unspliced xbp-1 mRNA levels in WT and *osm-3(dh441)IV* animals after 2 hr of 10 mM DTT treatment (*Figure 2F*).

**Figure supplement 1.** Quantitative PCR measuring relative mRNA expression of indicated genes in WT animals and *osm-3(dh441)IV* mutants after 6 hr of vehicle or TM treatment.

**Figure supplement 1—source data 1.** Quantitative PCR measuring relative mRNA expression of indicated genes in WT animals and *osm-3(dh441)IV* mutants after 6 hr of vehicle or TM treatment.

treatment (*Figure 3—figure supplement 1*, *Figure 3—figure supplement 1—source data 1*). Given the links between the insulin signaling pathway and ER stress signaling (*Henis-Korenblit et al., 2010*; *Kyriakakis et al., 2017*; *Matai et al., 2019*; *Labbadia and Morimoto, 2014*), we were surprised to find that *daf-2* mutants failed to develop on 10 µg/mL TM, while *osm-3* mutants were fully developed (*Figure 3—figure supplement 1*, *Figure 3—figure supplement 1—source data 2*). Together, this evidence suggests that the *osm-3* lifespan extension and TM drug resistance are uncoupled and act via independent pathways.

In *C. elegans*, *Pseudomonas aeruginosa* PA14 is a pathogen that is frequently used to study innate immunity. The PMK-1 p38 mitogen-activated protein kinase (MAPK) pathway as well as UPR[ER] targets have been implicated in the innate immune response of *C. elegans* during PA14 infection (*Richardson et al., 2010*; *Haskins et al., 2008*). Given the role of ER homeostasis in innate immunity and the TM resistance of *osm-3* mutants, we speculated that they would be more robust on PA14 than WT. Indeed, *osm-3* mutants were resistant to PA14, and this increased pathogen resistance was lost upon *pmk-1* knockdown (*Figure 3C*). With this in mind, we tested whether *pmk-1* mutation could suppress the *osm-3* TM resistance. The *osm-3; pmk-1* double mutant was fully resistant to TM, while the *pmk-1* single mutant was sensitive (*Figure 3D*, *Figure 3—source data 2*), demonstrating that TM resistance and *pmk-1* mediated pathogen resistance are uncoupled in *osm-3* mutants. Taken together, insulin signaling and the MAPK pathway, two major stress response pathways in *C. elegans* that have links to ER protein quality control, are not required for TM resistance in *osm-3* mutants.

## NHR-8 and PGPs regulate xenobiotic detoxification and TM resistance in *osm-3* mutants

Given the absence of ER stress in TM-treated *osm-3* mutants along with the genetic separation of TM resistance from insulin or MAPK signaling, we hypothesized that TM might be cleared from worm tissues through xenobiotic detoxification. PGPs are a conserved family of ATP binding cassette (ABC) transporters that locate to the cell membrane (*Sangster, 1994*) and they are involved in toxin clearance. *C. elegans* have 15 *pgp* genes. Using quantitative PCR, we found a significant upregulation of *pgp-3*, *pgp-5*, *pgp-11*, and *pgp-13* in *osm-3* mutants (*Figure 4A*). The nuclear hormone receptor NHR-8 has been linked to xenobiotic detoxification and *pgp* regulation in *C. elegans* (*Ménez et al., 2019*; *Lindblom et al., 2001*). Indeed, we found that the upregulation of *pgp-3*, *pgp-5*, *pgp-11* and *pgp-13*, in *osm-3* mutants, was suppressed in the *osm-3; nhr-8* double mutant (*Figure 4A*, *Figure 4—source data*

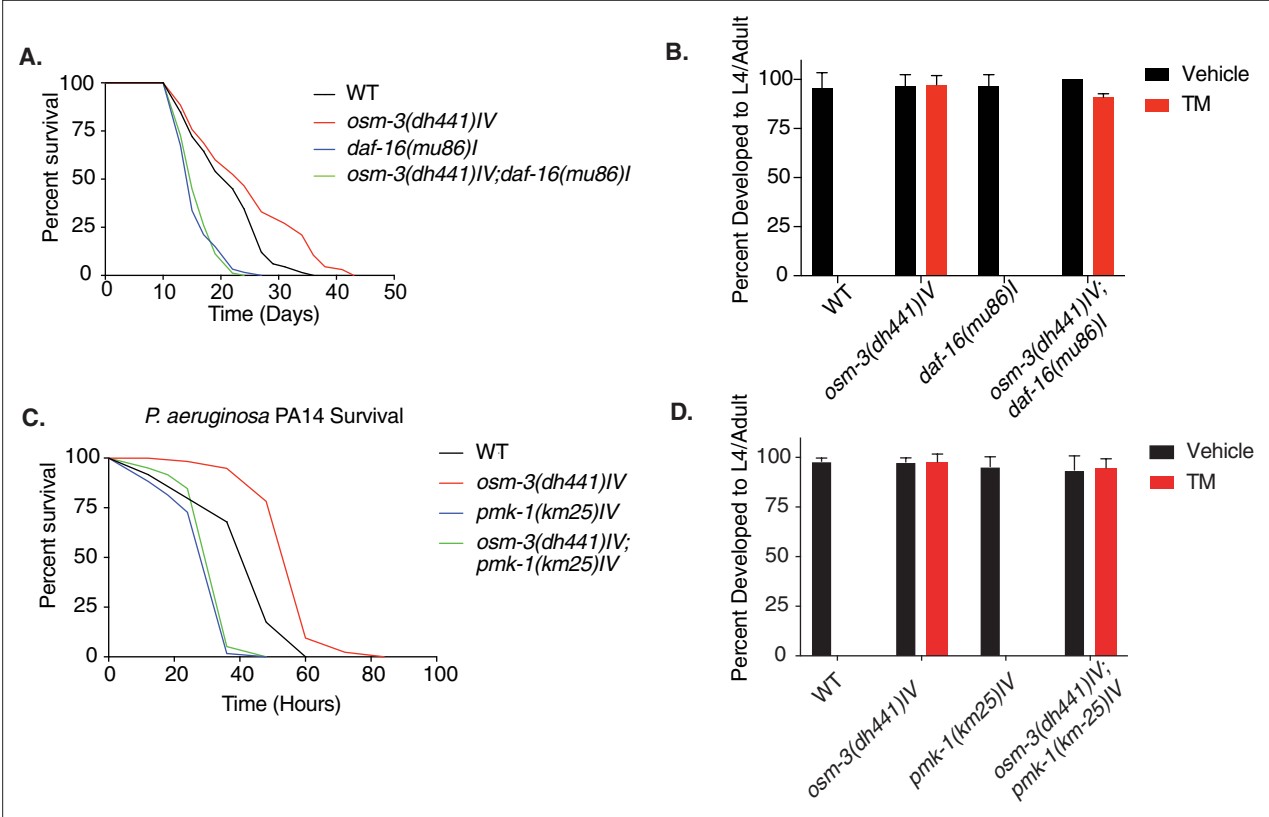

**Figure 3.** Tunicamycin resistance in *osm-3* mutants is not *daf-16* or *pmk-1* dependent. (**A**) Demographic lifespan analysis of WT, *osm-3(dhh441)IV*, *daf-16(mu86)I* and *osm-3(dh441)IV; daf-16(mu86)I* animals. WT mean lifespan = 22 days, *osm-3(dh441)IV* mean lifespan = 25 days p<0.005 compared to WT, *daf-16(mu86)I* mean lifespan = 16 days p<0.0001 compared to WT, *osm-3(dh441)IV; daf-16(mu86)I* mean lifespan = 16 days p<0.0001 compared to WT. See ***Supplementary file 1*** for statistical analysis. (**B**) Developmental resistance assay using 10 µg/mL TM with WT, *osm-3(dh441) IV*, *daf-16(mu86) I*, and *osm-3(dh441)IV; daf-16(mu86)I* animals. No viable WT or *daf-16(mu86)* animals were observed in the TM condition. Data are mean ± SD (n = 3 independent experiments with >15 scored animals each). (**C**) *Pseudomonas aeruginosa* PA14 survival assay with WT, *osm-3(dh441) IV*, *pmk-1(km25)IV*, and *osm-3(dh441)IV; pmk-1(km25)IV* animals. WT mean survival = 44 hr, *osm-3(dh441)IV* mean survival = 58 hr p<0.001 compared to WT, *pmk-1(km25)IV* mean survival = 31 hr p<0.001 compared to WT, *osm-3(dh441)IV; pmk-1(km25)IV* mean survival = 34 hr p<0.001 compared to WT and p=0.06 compared to *pmk-1(km25)IV*. See ***Supplementary file 1*** for statistical analysis. (**D**) Developmental resistance assay using 10 µg/mL TM with WT, *osm-3(dh441)IV*, *pmk-1(km25)IV*, and *osm-3(dh441)IV; pmk-1(km25)IV* animals. No viable WT or *pmk-1(km-25)* animals were observed in the TM condition. Data are mean ± SD (n = 3 independent experiments with >15 scored animals each).

The online version of this article includes the following figure supplement(s) for figure 3:

**Source data 1.** Developmental resistance assay using 10 µg/mL TM with WT, *osm-3(dh441)IV, daf-16(mu86)I*, and *osm-3(dh441)IV; daf-16(mu86)I* mutants.

**Source data 2.** Developmental resistance assay using 10 µg/mL TM with WT, *osm-3(dh441)IV, pmk-1(km25)IV*, and *osm-3(dh441)IV; pmk-1(km25)IV* mutants.

**Figure supplement 1.** Tunicamycin resistance in *osm-3* mutants is not *daf-16* or *pmk-1* dependent.

**Figure supplement 1—source data 1.** Biosorter analysis of *osm-3(dh441)IV* vs. WT animals in the *hsp-4*::GFP background raised on control or *daf-16* RNAi.

**Figure supplement 1—source data 2.** Developmental resistance assay using 10 µg/mL TM with WT, *osm-3(dh441) IV,* and *daf-2(e1370) III* mutants.

*1*). PGP upregulation suggested a role for PGPs in the TM resistance of *osm-3* mutants. Therefore, we tested if RNAi-mediated PGP knockdown in *osm-3* mutants would suppress TM resistance. We focused on the upregulated *pgp-5, pgp-11* and *pgp-13*, and on *pgp-8* and *pgp-12*. Interestingly, knockdown of *pgp-5, pgp-11*, and *pgp-12* partially reduced the TM resistance of the *osm-3* mutants (***Figure 4B***, ***Figure 4—source data 2***). Next, we used verapamil to inhibit PGPs. Verapamil has been used to specifically inhibit PGP activity, re-sensitizing worms to the anthelmintic compound ivermectin (***Ménez et al., 2016***). Indeed, 1 nM verapamil significantly suppressed development of *osm-3 C. elegans* in the presence of TM, showing no effect on controls without TM (***Figure 4C***, ***Figure 4—source data 3***).

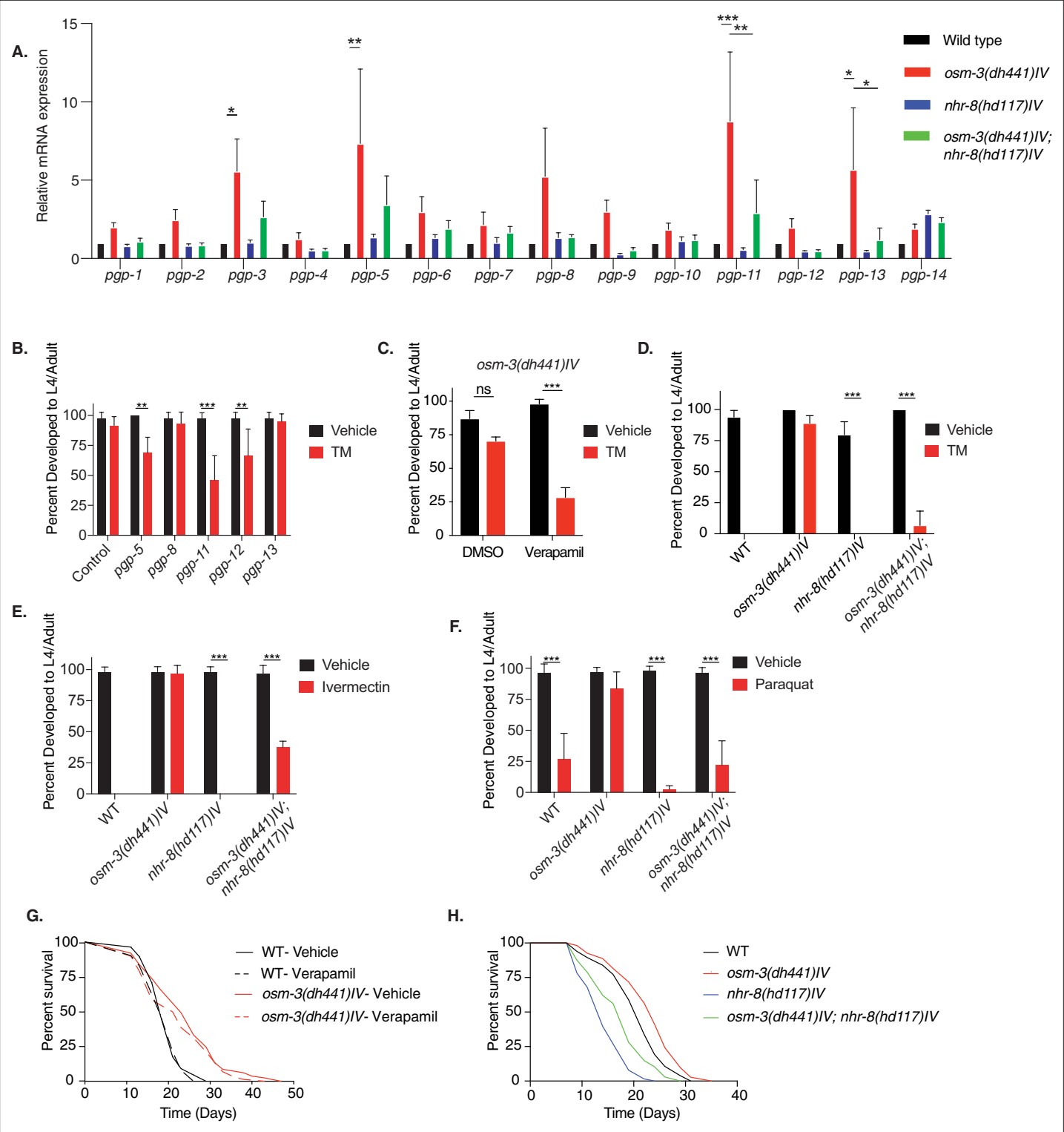

**Figure 4.** *nhr-8* signaling regulates xenobiotic detoxification response through PGPs. (**A**) Quantitative PCR measuring relative PGP mRNA expression in WT, *osm-3(dh441)IV*, *nhr-8(hd117)IV*, and *osm-3(dh441)IV; nhr-8(hd117)IV* animals. Relative mRNA expression is mRNA expression levels normalized to WT. There was no TM treatment. Data are mean + SEM, n = 3, *p<0.05, **p<0.001, ***p<0.0001 by two-way ANOVA. (**B**) Developmental TM resistance assay using 10 µg/mL TM and the indicated RNAi treatments with *osm-3(dh441)IV* mutants. Data are mean + SD, n = 3, **p<0.05, ***p<0.001 by two-way ANOVA. (**C**) Developmental TM resistance assay using the PGP inhibitor verapamil (VPL) using 10 µg/mL TM supplemented with vehicle or 1 nM VPL. Data are mean + SD, n = 3, ***p<0.001 by two-way ANOVA. (**D**) Developmental TM resistance assay on 10 µg/mL TM and control with WT, *osm-3(dh441)IV*, *nhr-8(hd117)IV*, and *osm-3(dh441)IV; nhr-8(hd117)IV* animals. Data are mean + SD, n = 3, *p<0.05, ***p<0.0001 by t-test. (**E**) Developmental

*Figure 4 continued on next page*

*Figure 4 continued*

ivermectin resistance assay of WT, *osm-3(dh441)IV*, *nhr-8(hd117)IV*, and *osm-3(dh441)IV; nhr-8(hd117)IV* animals using 6 ng/mL ivermectin and vehicle control. Data are mean + SD, n = 3, ***p<0.001 by two-way ANOVA. (**F**) Developmental paraquat resistance assay of WT, *osm-3(dh441)IV*, *nhr-8(hd117) IV*, and *osm-3(dh441)IV; nhr-8(hd117)IV* animals using 0.2 mM paraquat and vehicle control. Data are mean + SD, n = 3, ***p<0.001 by two-way ANOVA. (**G**) Demographic lifespan analysis on vehicle and verapamil-treated WT and *osm-3(dh441)IV* worms. Vehicle treated.WT mean lifespan = 19 days; *osm-3(dh441)IV* mean lifespan = 24 days p<0.0001 compared to WT vehicle. verapamil treated.WT mean lifespan = 19 days; *osm-3(dh441)IV* mean lifespan = 22 days, p<0.005 compared to WT Vehicle. (**H**) Demographic lifespan analysis of WT, *osm-3(dh441)IV*, *nhr-8(hd117)IV*, and *osm-3(dh441)IV;nhr-8(hd117)IV* animals. WT mean lifespan = 22 days, *osm-3(dh441)IV* mean lifespan = 24 days p<0.0001 compared to WT, *nhr-8(hd117)IV* mean lifespan = *14* p<0.0001 compared to WT, *osm-3(dh441)IV; nhr-8(hd117)IV* mean lifespan = 19 days p<0.0001 compared to WT.

The online version of this article includes the following figure supplement(s) for figure 4:

**Source data 1.** Quantitative PCR measuring relative PGP mRNA expression in WT, *osm-3(dh441)IV*, *nhr-8(hd117)IV*, and *osm-3(dh441)IV; nhr-8(hd117)IV* animals (*Figure 4A*).

**Source data 2.** Developmental TM resistance assay using 10 µg/mL TM and the indicated PGPs RNAi treatment.

**Source data 3.** Developmental TM resistance assay using the PGP inhibitor verapamil (VPL) using 10 µg/mL TM supplemented with vehicle or 1 nM VPL.

**Source data 4.** Developmental TM resistance assay on 10 µg/mL TM and control with WT, *osm-3(dh441)IV*, *nhr-8(hd117)IV* and *osm-3(dh441)IV; nhr-8(hd117)IV* animals (*Figure 4D*).

**Source data 5.** Developmental ivermectin resistance assay of WT, *osm-3(dh441)IV*, *nhr-8(hd117)IV* and *osm-3(dh441)IV; nhr-8(hd117)IV* animals using 6 ng/mL ivermectin and vehicle control (*Figure 4E*).

**Source data 6.** Developmental paraquat resistance assay of WT, *osm-3(dh441)IV*, *nhr-8(hd117)IV*, and *osm-3(dh441)IV; nhr-8(hd117)IV* animals using 0.2 mM paraquat and vehicle control (*Figure 4F*).

**Figure supplement 1.** *nhr-8* signaling regulates xenobiotic detoxification response through PGPs.

**Figure supplement 1—source data 1.** Heat shock assay at 35 °C in WT and *osm-3(dh441)IV* animals on day 1 of adulthood.

**Figure supplement 1—source data 2.** Hydrogen peroxide survival assay using 1 mM H2O2 on day 1 of adulthood in WT and *osm-3(dh441)IV* animals.

**Figure supplement 1—source data 3.** WT development on 0–1 µM Verapamil.

The partial effects of *pgp* knockdown (*Figure 4B*) and verapamil treatment (*Figure 4C*) suggest that xenobiotic detoxification is achieved by multiple PGPs, which are likely to be transcriptionally orchestrated by *nhr-8*. This led us to ask whether *osm-3* TM resistance was dependent on NHR-8. Strikingly, the developmental TM assay revealed that the resistance of *osm-3* mutants was largely suppressed in the *osm-3; nhr-8(hd117)IV* double mutant (*Figure 4D*. *Figure 4—source data 4*), suggesting that NHR-8 is necessary for *osm-3* TM resistance. As suggested by a role of PGPs in ivermectin resistance, we found that *osm-3* mutants fully developed to adults in the presence of 6 µg/mL ivermectin that is lethal to WT animals (*Figure 4E*, *Figure 4—source data 5*). Ivermectin resistance of *osm-3* mutants was suppressed by loss of NHR-8. Dyf *C. elegans* mutants have been independently identified in drug resistance screens (*Fujii et al., 2004*; *Ménez et al., 2016*; *Dent et al., 2000*). Consistent with these findings, we observed that *osm-3* mutants were resistant to 200 nM paraquat, as they fully develop at a concentration that is toxic to WT *C. elegans* (*Figure 4F*, *Figure 4—source data 6*). Consistent with a role in xenobiotic detoxification, NHR-8 was required for the paraquat resistance of the *osm-3* mutants (*Figure 4F*).

We next reasoned that resistance of *osm-3* mutants should be limited to chemical stressors that can be eliminated by xenobiotic detoxification. We thus tested heat stress and hydrogen peroxide (H₂O₂) induced oxidative stress. Paraquat and hydrogen peroxide are both oxidative stressors, but paraquat acts indirectly via the mitochondria (*Castello et al., 2007*), while hydrogen peroxide is a primary cause for oxidative damage. When challenged with heat or hydrogen peroxide, *osm-3* mutants showed no difference to WT animals, suggesting that *osm-3* mutants are not resistant to these non-xenobiotic stressors (*Figure 4—figure supplement 1A, B*, *Figure 4—figure supplement 1—source data 1* and *Figure 4—figure supplement 1—source data 2*).

Given the involvement of PGP-dependent xenobiotic detoxification in TM resistance of *osm-3* mutants, we wondered about its role in the longevity of *osm-3* mutants. Notably, verapamil did not shorten *osm-3* lifespan (*Figure 4G*). Moreover, verapamil did not affect WT lifespan (*Figure 4G* and *Supplementary file 1*) and had no visible effect on WT development (*Figure 4—figure supplement 1C*, *Figure 4—figure supplement 1—source data 3*). Finally, we tested a possible role of NHR-8 in *osm-3* longevity. While *nhr-8* mutation shortens lifespan, the *osm-3* mutation extended both WT and *nhr-8* survival to a similar degree (*Figure 4H*). Together, these data support the conclusion that the

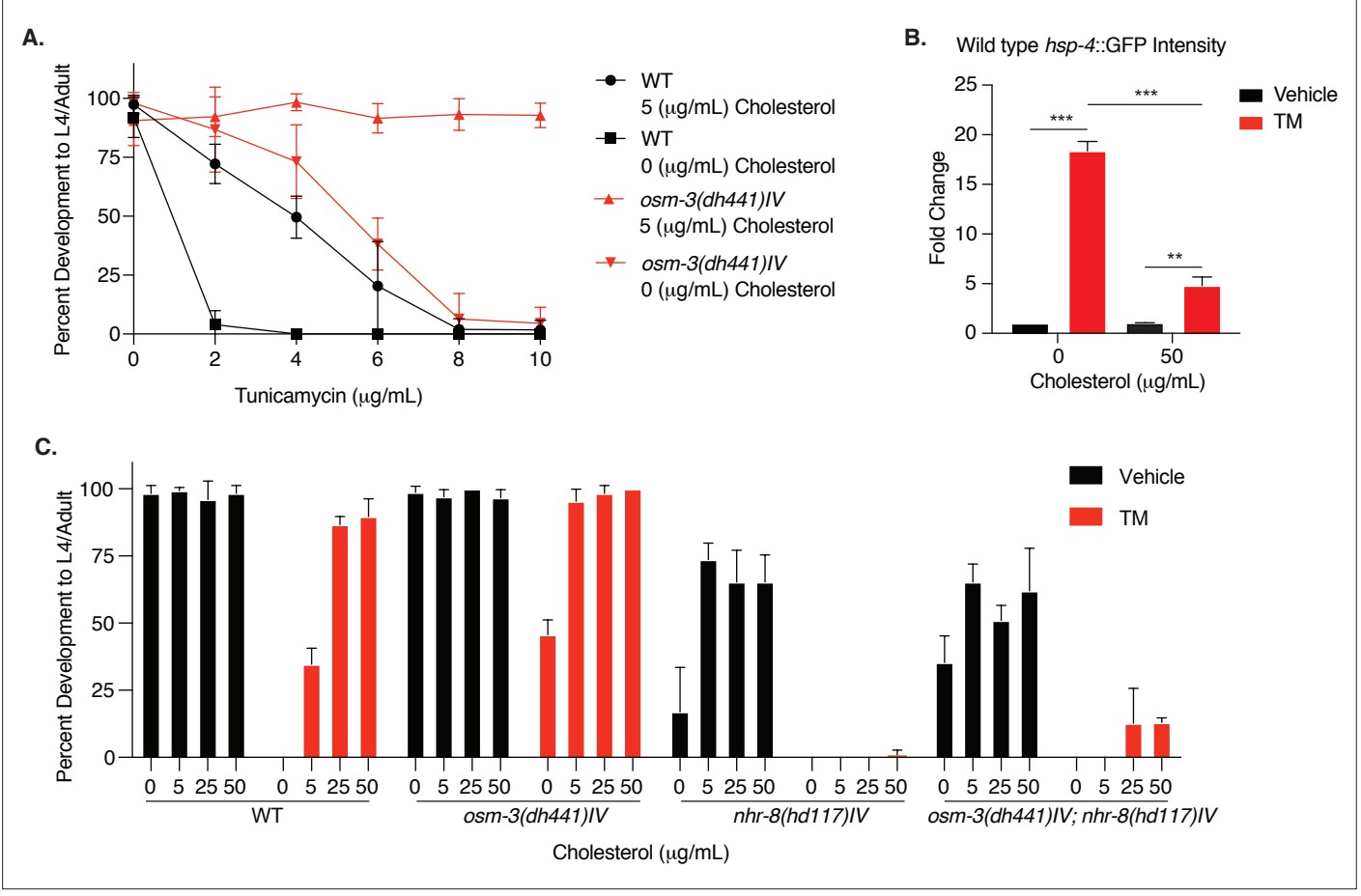

**Figure 5.** Cholesterol modulates TM resistance through NHR-8. (**A**) Developmental dose–response TM resistance assay using the indicated TM concentrations with WT animals and *osm-3(dh441)IV* mutants raised on 0 or 5 µg/mL cholesterol. Data are mean ± SD (n = 5 independent experiments with >15 scored animals each). (**B**) Biosorter analysis of WT animals with the *hsp-4*::GFP reporter raised on 0 or 50 µg/mL cholesterol after 6 hr of vehicle or 5 µg/mL TM treatment. Data are mean + SEM, n = 3, *p<0.05, ***p<0.0001 by two-way ANOVA. (**C**) Cholesterol dose–response developmental assay with WT, *osm-3(dh441)IV*, *nhr-8(hd117)IV*, and *osm-3(dh441)IV; nhr-8(hd117)IV* animals on vehicle or 5 µg/mL TM. Data are mean ± SD (n = 3 independent experiments with >15 scored animals each).

The online version of this article includes the following figure supplement(s) for figure 5:

**Source data 1.** Developmental dose–response TM resistance assay using the indicated TM concentrations with WT animals and *osm-3(dh441)IV* mutants raised on 0 or 5 µg/mL cholesterol.

**Source data 2.** Biosorter analysis of WT animals with the hsp-4::GFP reporter raised on 0 or 50 µg/mL cholesterol after 6 hr of vehicle or 5 µg/mL TM treatment.

**Source data 3.** Cholesterol dose–response developmental assay with WT, *osm-3(dh441)IV*, *nhr-8(hd117)IV*, and *osm-3(dh441)IV; nhr-8(hd117)IV* animals on vehicle or 5 µg/mL TM.

xenobiotic stress resistance and longevity phenotypes of *osm-3* mutants are mediated by independent pathways.

## Sterol signals modulate TM resistance

We aimed to further understand the systemic signals that control xenobiotic detoxification of TM. The NHR-8 ligand remains unknown, but is likely a sterol molecule derived from cholesterol metabolism (**Magner et al., 2013**). We thus tested the role of cholesterol in TM resistance and found that omitting cholesterol from the culture plates for one generation leads to TM hypersensitivity in both WT and *osm-3* animals (**Figure 5A**, **Figure 5—source data 1**). To specifically assess a role of cholesterol-derived signals in TM resistance, we used the *hsp-4*::GFP reporter and found that high cholesterol significantly reduced TM-induced ER stress (**Figure 5B**, **Figure 5—source data 2**). Developmental

cholesterol dose–response assays in the presence of TM confirmed that WT TM resistance is cholesterol dependent. In *nhr-8* and *osm-3; nhr-8* double mutants, however, cholesterol treatment failed to protect from TM toxicity (*Figure 5C*, *Figure 5—source data 3*). This demonstrates a key role of NHR-8 in mediating TM resistance downstream of sterol signals.

## Discussion

In this study, we found that *osm-3* mutants as well as other long-lived Dyf mutants are resistant to the ER toxin TM. Instead of an activated stress signaling status that might explain the TM resistance, *osm-3* Dyf mutants show no UPR$^{ER}$ induction upon TM treatment. Consistent with previous findings (*Apfeld and Kenyon, 1999*), the lifespan extension of *osm-3* was *daf-16* dependent. Despite the established link between the insulin signaling pathway and stress resistance, the TM resistance in *osm-3* mutants was not insulin signaling dependent. Furthermore, *osm-3* mutants were pathogen resistant, and this resistance was fully dependent on the PMK-1/p38 MAPK pathway. We were surprised that neither of the two pathways weakened the TM resistance phenotype. Dyf *C. elegans* drug resistance is not specific to TM, as we also observed resistance to paraquat and ivermectin. While paraquat and ivermectin resistance have been previously reported in Dyf mutants (*Dent et al., 2000*; *Fujii et al., 2004*; *Ménez et al., 2016*), no studies have demonstrated similar resistance mechanisms in Dyf *C. elegans* mutants to TM. Our findings link drug resistance to increased PGP expression in Dyf worms, suggesting a causal role of xenobiotic detoxification in the drug resistance phenotype. Importantly, we found that the regulation of a subset of PGPs is mediated by the nuclear hormone receptor NHR-8. Genetic or pharmacological PGP inhibition, as well as loss of *nhr-8* significantly resensitized *osm-3* mutants to TM, ivermectin, and paraquat. Potential NHR-8 ligands are derived from cholesterol (*Magner et al., 2013*), and we observed that WT development on lower TM concentrations was strongly cholesterol and NHR-8 dependent.

Given that improved protein homeostasis is one of the cellular hallmarks of longevity (*López-Otín et al., 2013*), we presumed that selection for TM resistance would serve as a proxy phenotype for longevity given that TM specifically targets ER protein folding. More specifically, we expected that high fidelity ER protein quality control would be the driver of longevity in these mutants. Previous findings describing increased hexosamine biosynthetic pathway flux (*Denzel et al., 2014*) or constitutive XBP-1 activation *Taylor and Dillin, 2013* have linked protein homeostasis to longevity and stress resistance (*Denzel and Antebi, 2015*); therefore, the connection between TM resistance and longevity seemed apparent. Given that Dyf mutants are resistant to pathogenic bacteria through enhanced ER homeostasis, we speculated that TM resistance might serve as a proxy phenotype for pathogen resistance. Contrary to that, we found that TM resistance is a consequence of xenobiotic detoxification via PGPs, and NHR-8 regulated control of PGPs was not involved in the longevity of *osm-3* mutants. *pmk-1,* and *daf-16* were required for pathogen resistance and longevity, respectively, but not for the drug resistance of *osm-3* mutants. Together, these findings demonstrate that the three phenotypes of *osm-3* mutants (i.e. xenobiotic resistance, longevity, and protection from pathogens) are independent and controlled by separable genetic pathways. Our observations thus support a model in which the chemosensation defect activates a number of parallel pathways that are responsible for the distinct phenotypes of *osm-3* mutants.

Adding another dimension to the cell-nonautonomous regulation of innate immunity (*Aballay, 2013*), we propose that the nervous system regulates systemic xenobiotic detoxification in *C. elegans*, likely through a sterol ligand. Our data suggest that mutants with defective amphid neurons have increased PGP expression. Several PGPs have been shown to be expressed in non-neuronal tissue (*Lincke et al., 1993*; *Sheps et al., 2004*). Moreover, we found that PGP expression is regulated by *nhr-8*, which has also been shown to be expressed in the intestine of *C. elegans* (*Magner et al., 2013*). Combining this previous information with our finding that Dyf mutants are resistant to several uniquely toxic drugs, we conclude that neuronal signaling controls drug resistance through nuclear hormone signaling. Based on our results, we cannot make conclusions about the sterol metabolism involved in the production of a putative NHR-8 ligand and we cannot make direct conclusions about PGP regulation by sterols. Given our observation that supplementation with cholesterol is itself sufficient for TM resistance in WT worms, it is now important to investigate the link between neuronal states and the specific cholesterol-derived signal that drives these drug resistance phenotypes.

PGPs are likely regulated in *C. elegans* to help them combat toxins found in their natural habitat. Toxic metabolites are undoubtedly common in the soil where *C. elegans* are found. The drugs used in our study, ivermectin (a derivative of avermectin) (**Burg et al., 1979**) and TM, were first discovered as antibiotics that are produced by soil bacteria (**Takatsuki et al., 1971**). Because *C. elegans* have no adaptive immune response, having PGPs as part of their innate immune system allows them to clear toxic molecules they may encounter in the wild. While Dyf mutants likely do not occur in the wild, our data nonetheless demonstrate a link between environmental sensing and drug resistance.

In humans, PGPs have been implicated in drug resistant malignancies (**Lehne, 2000**). Using *C. elegans* as a tool to study the cross talk between tissues, one might be able to better understand how extracellular signaling drives ABC transporter expression in chemotherapy-resistant cancer. Further studies in cell culture might also characterize TM, as well as other toxic metabolites, as substrates for *C. elegans* PGPs.

# Materials and methods

**Key resources table**

| Reagent type (species) or resource | Designation | Source or reference | Identifiers | Additional information |
|---|---|---|---|---|
| Strain, strain background (*E. coli*) | HT115 [*L4440::sel-1*] | Source Bioscience | CELE_F45D3.5 | Ahringer RNAi library |
| Strain, strain background (*E. coli*) | HT115 [*L4440::daf-16*] | Source Bioscience | CELE_R13H8.1 | Ahringer RNAi library |
| Strain, strain background (*E. coli*) | HT115 [*L4440::pgp-5*] | Source Bioscience | CELE_C05A9.1 | Ahringer RNAi library |
| Strain, strain background (*E. coli*) | HT115 [*L4440::pgp-8*] | Source Bioscience | CELE_T21E8.3 | Ahringer RNAi library |
| Strain, strain background (*E. coli*) | HT115 [*L4440::pgp11*] | Source Bioscience | CELE_DH11.3 | Ahringer RNAi library |
| Strain, strain background (*E. coli*) | HT115 [*L4440::pgp-12*] | Source Bioscience | CELE_F22E10.1 | Ahringer RNAi library |
| Strain, strain background (*E. coli*) | HT115 [*L4440::pgp-13*] | Source Bioscience | CELE_F22E10.2 | Ahringer RNAi library |
| Genetic reagent (*C. elegans*) | *osm-3(dh441)IV* | Other | AA1962 | available from A. Antebi or M.S. Denzel |
| Genetic reagent (*C. elegans*) | *daf-16(mu86)I* | CGC | CF1038 | |
| Genetic reagent (*C. elegans*) | *pmk-1(km25)IV* | CGC | KU25 | |
| Genetic reagent (*C. elegans*) | *osm-3(dh441)IV; nhr-8(hd117)IV* | This paper | MSD420 | available from M.S. Denzel |
| Genetic reagent (*C. elegans*) | *osm-3(dh441)IV; daf-16(mu86)I* | This paper | MSD422 | available from M.S. Denzel |
| Genetic reagent (*C. elegans*) | *osm-3(dh441)IV; pmk-1(km25)IV* | This paper | MSD423 | available from M.S. Denzel |
| Genetic reagent (*C. elegans*) | *odr-1(n1936)X* | CGC | CX2065 | |
| Genetic reagent (*C. elegans*) | *odr-3(n2150)V* | CGC | CX2205 | |
| Genetic reagent (*C. elegans*) | *osm-6(p811)V* | CGC | PR811 | |
| Genetic reagent (*C. elegans*) | *tax-4(p678)III* | CGC | PR678 | |

*Continued on next page*

*Continued*

| Reagent type (species) or resource | Designation | Source or reference | Identifiers | Additional information |
|---|---|---|---|---|
| Genetic reagent (*C. elegans*) | *nhr-8(hd117)IV* | other | AA968 | available from A. Antebi or M.S. Denzel |
| Genetic reagent (*C. elegans*) | *N2; dhEx451(nhr-8::gfp; coel::RFP)* | CGC | AA1027 | |
| Genetic reagent (*C. elegans*) | *osm-3(dh441)IV; zcIs4 [hsp-4::GFP]V* | other | AA2774 | available from A. Antebi or M.S. Denzel |
| Genetic reagent (*C. elegans*) | *zcIs4[hsp-4::GFP]V* | CGC | SJ4005 | |
| Genetic reagent (*C. elegans*) | *che-11(e1810)V* | CGC | CB3330 | |
| Genetic reagent (*C. elegans*) | *daf-10(e1387)IV* | CGC | CB1387 | |
| Genetic reagent (*C. elegans*) | *osm-1(p808)X* | CGC | PR808 | |
| Genetic reagent (*C. elegans*) | *dyf-1(mn335)I* | CGC | SP1205 | |
| Genetic reagent (*C. elegans*) | *che-2(e1033)X* | CGC | CB1033 | |
| Genetic reagent (*C. elegans*) | *osm-5(p813)X* | CGC | PR813 | |
| Commercial assay or kit | Direct-zol RNA Microprep | Zymo Research | R2060 | |
| Commercial assay or kit | SMARTer Stranded RNA-Seq HT Kit | Takara | 634,838 | |
| Commercial assay or kit | Ribo-zero Gold kit | Illumina | MRZG12324 | |
| Chemical compound, drug | Tunicamycin | Calbiochem | CAS 11089-65-9 | |
| Chemical compound, drug | Dithiothreitol | Sigma-Aldrich | CAS 3483-12-3 | |
| Chemical compound, drug | Methyl viologen dichloride hydrate | Sigma-Aldrich | CAS 75365-73-0 | Paraquat |
| Chemical compound, drug | Ivermectin | Sigma-Aldrich | CAS 70288-86-7 | |
| Chemical compound, drug | Verapamil hydrochloride | Sigma-Aldrich | CAS 152-11-4 | |
| Chemical compound, drug | Cholesterol | Sigma-Aldrich | CAS 57-88-5 | |
| Chemical compound, drug | 1,1′-Dioctadecyl-3,3,3′,3′-tetramethylindocarbo cyanine perchlorate | Sigma-Aldrich | CAS 41085-99-8 | DiI |
| Software, algorithm | R statistical software | cran.r-project.org | DESeq2 1.16.1 | PMID.25516281 |
| Other | Reference Genome | ENSEMBL | WBCel235 | |

## Worm maintenance

*C. elegans* nematodes were cultured using standard husbandry methods at 20 °C on nematode growth media (NGM) agar plates seeded with *E. coli* strain OP50 unless otherwise stated (*Brenner, 1974*).

## Dye filling assay in *C. elegans*

The dye filling assay was performed on a synchronized population of day one adults. Forty to 60 adult worms were placed in M9 containing 10 µg/mL DiI. They were left at room temperature for 2 hr in the staining medium, then transferred to NGM plates seeded with *E. coli* OP50. The worms were then allowed to crawl on the food for 1 hr to allow for the dye to be cleared from the gut. The worms were scored for dye filling using a Leica SPX-8 confocal microscope or a Leica fluorescent stereo-microscope. Images were taken using the confocal microscope.

## Drug resistance assays

NGM plates containing either TM, paraquat, or ivermectin were used in developmental assays to test for resistance. To this end, we performed a 4 hr egg lay on NGM plates at room temperature, then transferred the eggs to control and drug containing plates, and recorded the number of eggs per plate. After incubation for 4 days at 20 °C, plates were scored by counting the total number of L4 larvae or adults on each plate. Paraquat plates were prepared by adding 1 M methyl viologen dichloride (paraquat) directly onto seeded NGM plates for a final concentration of 200 µM and allowed to dry immediately before use. Ivermectin plates were prepared by adding ivermectin directly to the NGM medium at a final concentration of 6 µg/mL before pouring the plates. Modified NGM plates containing no cholesterol were made by using standard NGM without the addition of cholesterol. NGM agar plates containing 5, 25, or 50 µg/mL Cholesterol were used for development when specifically stated. These modified NGM plates were then supplemented with drugs as described above. All of the drug development assays were performed using *E. coli* OP50 or *E. coli* HT115 bacteria. For the TM RNAi experiments (*Figure 4B*), animals were cultured for one generation on *pgp-5*, *pgp-8*, *pgp-11*, *pgp-12*, or *pgp-13* RNAi supplemented with DMSO. The offspring was placed on RNAi plates supplemented with 10 µg/mL TM.

## ER stress quantification

Synchronized day one adults were transferred to control and TM containing NGM plates seeded with OP50. After 6 hr of stress induction, *hsp-4*::GFP levels were measured by large particle flow cytometry in both the WT and *osm-3(dh441)IV* background using the Union Biometrica Biosorter and imaged using a Leica fluorescent stereo-microscope. For RNAi experiments, animals were raised on control and *daf-16* or *sel-1* RNAi. Synchronized day one adults were transferred to vehicle or TM containing plates seeded with control or experimental RNAi. After 6 hr of stress, GFP levels were measured using the Biosorter. GFP values were normalized to time of flight. At least 100 animals were analyzed per condition. For RNA isolation, day one adults were treated with 10 mM DTT in S Basal buffer containing *E. coli* OP50 for 2 hr and then snap frozen.

## Quantitative PCR

For ER stress induction day one adults were washed from their plates and transferred to either control or TM containing plates, where they were incubated for 6 hr. The animals were then washed off using M9 and snap frozen in trizol. Unstressed synchronized animals were collected at day 1 of adult hood for RNA extraction. RNA was prepared using Zymo Research Direct-zol RNA Microprep kit. SYBR green was used to perform quantitative PCR (RT-qPCR). See *Supplementary file 2* for list of qPCR primers used in this study. Primer efficiency was tested and was always above 90 % and below 110%.

## Lifespan analysis

Adult lifespan analysis was performed at 20 °C on mutant and WT *C. elegans*. The animals were synchronized in a 4-hr egg lay. Animals were scored as dead or alive every second day until the end of the experiment. Animals were transferred every day for the first 7 days. Statistical analysis on the Kaplan–Meier survival curves was performed using Microsoft Excel. See *Supplementary file 1* for lifespan statistics. Each survival experiment was performed at least twice.

## Hydrogen peroxide survival assay

One molar hydrogen peroxide ($H_2O_2$) was added to unseeded NGM plates to a final concentration of 1 µM and allowed to dry for several minutes. Synchronized day one adults were then transferred onto the plates and incubated at 25 °C. The animals were then scored every 2 hr for survival. Twenty to 50 animals were used in each experimental condition.

## Pathogenic bacteria survival assay

The *P. aeruginosa* strain PA14 was grown in LB media and seeded onto high peptone NGM plates and incubated for 12 hr at 37 °C immediately before the start of the experiment. Day one animals were first transferred to unseeded NGM plates to crawl around the plate and remove any excess *E. coli* OP50 off of their bodies. They were then transferred to the PA14 containing plates and incubated at

25 °C. The animals were scored every 12 hr until all animals were dead. At least 75 animals were used in each experimental condition.

### Heat stress survival assay

Day one synchronized *C. elegans* were transferred to fresh NGM plates seeded with *E. coli* OP50. These plates were transferred to a 35 °C incubator, where they were evenly distributed throughout the incubator to ensure even heat exposure. The plates were scored for live worms every 2 hr until all of the worms were dead. Twenty to 50 animals were used in each experimental condition.

### Sample collection and RNA purification for sequencing

For RNA sequencing, we used day one adults that were hatched within 1 hr. To achieve this synchronization, we washed all adults and larvae of plates with M9 and allowed the eggs on hatch for 1 hr. The freshly hatched L1 larvae were then washed off and transferred to fresh plates seeded with OP50 and incubated at 20 °C until they developed to adults. At day 1 of adulthood, animals were washed from their plates and transferred to either control or TM containing plates, where they were incubated for 6 hr. The animals were then washed off using M9 and snap frozen in trizol. All biological replicates were collected and prepared using a snaking collection method to reduce batch effects. Total RNA was purified using Zymo Research Direct-zol RNA Microprep kit.

### RNA-Seq library preparation

RNA quality was assessed using Agilent's Bioanalyzer platform, and samples with RIN >9 were used for library construction. 2.5 µg of total RNA was subjected to ribosomal RNA (rRNA) depletion using the Illumina's Ribo-zero Gold kit (Illumina), according to the manufacturer's protocol. Strand specific RNA-seq libraries were then constructed using the SMARTer Stranded RNA-Seq HT Kit (Clontech #634839), according to the manufacturer's protocol. Based on rRNA-depleted input amount, 15 cycles of amplification were performed to generate RNA-seq libraries. Paired-end 150 bp reads were sent for sequencing on the Illumina HiSeq-Xten platform at the Novogene Corporation (USA). The resulting data was then analyzed with a standardized RNA-seq data analysis pipeline (described below).

RNA-seq analysis pipeline cDNA sequences of worm genes were obtained through ENSEMBL Biomart for the WBCel235 build of the *C. elegans* genome (Ensemble release v94; accessed 2018-12-03). Trimmed reads were mapped to this reference using kallisto 0.43.0–1 and the –fr-stranded option (*Bray et al., 2016*). All subsequent analyses were performed in the R statistical software (https://cran.r-project.org/). Read counts were imported into R and summarized at the gene level. Differential gene expression was estimated using the 'DESeq2' R package (DESeq2 1.16.1). DAVID analysis (version 6.8) was performed to identify significantly enriched GO terms.

### Developmental verapamil resistance assay and lifespan analysis

Plates containing 10 µg/mL TM were supplemented with 1 nM verapamil by adding verapamil solved in DMSO onto the plates. Plates were allowed to dry for 6 hr before eggs were transferred to them to determine developmental resistance to TM. Plates were then scored after 4 days to determine the relative number of L4 larvae or adults. Lifespan assays were performed as above; however, the animals were transferred to fresh verapamil containing plates every second day for the whole lifespan assay. Twenty to 50 animals were used in each experimental condition.

## Acknowledgements

We thank all Denzel lab members for lively and helpful discussions. We thank Adam Antebi for valuable comments and for *C. elegans* strains. We thank Seung-Jae Lee for valuable comments on the manuscript. We thank the *Caenorhabditis* Genetics Center (CGC) for strains. We thank the members of the bioinformatics core facility and the FACS and imaging core facility at MPI AGE. BAB was supported by a seed grant from the NAVIGAGE foundation, a generous gift from the Hanson-Thorell Family and the Kathleen Gilmore award in aging biology. MSD was supported by ERC-StG 640254, by the Deutsche Forschungsgemeinschaft (DFG, German Research Foundation) – Projektnummer 73111208 – SFB 829, and by the Max Planck Society.

## Additional information

### Funding

| Funder | Grant reference number | Author |
|---|---|---|
| H2020 European Research Council | ERC-StG 640254 | Martin S Denzel |
| Max-Planck-Gesellschaft | | Gabriel A Guerrero |
| German Research Foundation | 73111208 | Martin S Denzel |
| German Research Foundation | SFB 829 | Martin S Denzel |

The funders had no role in study design, data collection and interpretation, or the decision to submit the work for publication.

### Author contributions

Gabriel A Guerrero, Conceptualization, Data curation, Formal analysis, Investigation, Methodology, Validation, Visualization, Writing – original draft; Maxime J Derisbourg, Conceptualization, Formal analysis, Investigation, Project administration, Supervision, Validation, Visualization, Writing – review and editing; Felix AMC Mayr, Data curation, Formal analysis, Investigation; Laura E Wester, Marco Giorda, J Eike Dinort, Klara Schilling, María José Alonso-De Gennaro, Investigation; Matías D Hartman, Investigation, Supervision; Ryan J Lu, Data curation, Formal analysis, Investigation, Software; Bérénice A Benayoun, Data curation, Formal analysis, Software, Supervision; Martin S Denzel, Conceptualization, Funding acquisition, Investigation, Methodology, Project administration, Supervision, Validation, Writing – original draft, Writing – review and editing

### Author ORCIDs

Maxime J Derisbourg (ID) http://orcid.org/0000-0002-9293-1413
Bérénice A Benayoun (ID) http://orcid.org/0000-0002-7401-4777
Martin S Denzel (ID) http://orcid.org/0000-0002-5691-3349

### Decision letter and Author response

Decision letter https://doi.org/10.7554/eLife.53174.sa1
Author response https://doi.org/10.7554/eLife.53174.sa2

## Additional files

### Supplementary files

• Supplementary file 1. Table containing the statistical analysis of the lifespan, PA14, heat shock and H2O2 resistance assays.

• Supplementary file 2. List of qPCR primers used in this study.

• Transparent reporting form

### Data availability

Raw sequencing data were deposited to the NCBI Gene Expression Omnibus (GEO) under the accession number GSE144675.

The following dataset was generated:

| Author(s) | Year | Dataset title | Dataset URL | Database and Identifier |
|---|---|---|---|---|
| Denzel MS, Benayoun BA, Guerrero GA | 2020 | NHR-8 regulated P-glycoproteins uncouple xenobiotic stress resistance from longevity in chemosensory *C. elegans* mutants | https://www.ncbi.nlm.nih.gov/geo/query/acc.cgi?acc=GSE144675 | NCBI Gene Expression Omnibus, GSE144675 |

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
