## [Decision Letter]

**Acceptance summary:**

In this manuscript, Guerrero et al. characterize the interrelatedness of three different phenotypes -lifespan extension, resistance to the ER stress-inducing drug tunicamycin (TM) and enhanced immunity- in a chemosensory defective mutant, osm-3(dh441). Using this mutant, the authors made the discovery that TM resistance can be entirely uncoupled from increased longevity and bacterial pathogen resistance, which depend on DAF-16/FOXO transcription factor and PMK-1/ p38 MAP kinase, respectively. Instead, the authors found that a subset of P-glycoproteins (PGPs) are moderately upregulated in the osm-3 mutants, leading them to hypothesize that PGPs serve as an efflux pump for TM, thereby conferring endoplasmic reticulum (ER) stress resistance. Consistently, the authors found that the ER unfolded protein response (UPR) is dampened in the osm-3 mutants upon treatment with TM, even though the osm-3 mutants are competent for UPR induction by other means. In addition, knockdown of select pgps and treatment with the drug verapamil, a known inhibitor of PGPs, reduces TM resistance in osm-3 mutants, without an effect on longevity, supporting separate pathways for xenobiotic resistance and longevity. Finally, the authors provide evidence that a subset of the PGPs are regulated by the nuclear hormone receptor NHR-8 and mutations in this receptor render osm-3 mutants sensitive to TM and other xenobiotics. Overall this study provides useful insights into a phenomenon that has long been observed: the multiple stress-resistance phenotypes and extended longevity of worms with neuronal defects. In demonstrating that these different phenotypes are separable, the authors make it clear that independent pathways function to mediate xenobiotic resistance, innate immunity and longevity in a chemosensory mutant.

**Decision letter after peer review:**

Thank you for submitting your article "NHR-8 regulated P-glycoproteins uncouple xenobiotic stress resistance from longevity in chemosensory *C. elegans* mutants" for consideration by *eLife*. Your article has been reviewed by 4 peer reviewers, including Elçin Ünal as the Reviewing Editor and Reviewer #1, and the evaluation has been overseen by Vivek Malhotra as the Senior Editor.

The reviewers have discussed the reviews with one another and the Reviewing Editor has drafted this decision to help you prepare a revised submission.

Summary:

In this manuscript, Guerrero et al. characterize the interrelatedness of three different phenotypes -lifespan extension, resistance to the ER stress-inducing drug tunicamycin (Tm) and enhanced immunity in a chemosensory defective mutant, osm-3(dh441). Using this mutant, the authors made the discovery that Tm resistance can be entirely uncoupled from increased longevity and bacterial pathogen resistance, which depend on the DAF-16/FOXO transcription factor and PMK-1/ p38 MAP kinase, respectively. Instead, the authors found that a subset of P-glycoproteins (PGPs) are moderately upregulated in the osm-3 mutants, leading them to hypothesize that PGPs serve as an efflux pump for Tm, thereby conferring ER stress resistance. This hypothesis is consistent with the observation that the Tm-induced activation of the ER unfolded protein response (UPR) is not apparent in the osm-3 mutants. In addition, treatment with the drug verapamil abolishes the ER stress resistance in osm-3 mutants, supporting the involvement of PGPs. Finally, the authors provide evidence that PGP expression is mediated by a nuclear hormone receptor called NHR-8 and mutations in this receptor render osm-3 mutants sensitive to Tm, while overexpression of nhr-8 alone is sufficient to confer resistance. Therefore, the ER stress resistance in osm-3 mutants seems to rely upon a general signature of xenobiotic resistance, rather than intracellular stress response mechanisms such as UPR.

Overall this is a well-executed study that provides useful insights into a phenomenon that has long been observed: the multiple stress-resistance phenotypes and extended longevity of worms with neuronal defects. In demonstrating that these different phenotypes are separable, the authors make it clear that multiple downstream pathways act in parallel to mediate multi-stress resistance and longevity in a chemosensory mutant. However, the evidence supporting the importance of PGPs and their regulator NHR-8 for Tm resistance is somewhat lacking. The following experiments should be considered in order to fully support the key conclusion of this manuscript:

Essential revisions:

1) PGP involvement in Tm resistance: The importance of PGPs rests upon the observation that treatment with verapamil reduces animals' resistance to Tm. However, the specificity of verapamil is not demonstrated in the paper. This drug has been shown to block calcium channels, which may affect diverse processes. More generally, the degree of PGP overexpression in osm-3 is pretty mild: is it reasonable to conclude that degree of overexpression is sufficient to produce the observed Tm resistance? Another concern is the inconsistency between the RNA-seq and RT-qPCR data for pgp expression. For example, by RNA-seq, pgp-2, pgp-9 and pgp-14 are reported to have significant difference in expression (Figure 4A and 4B) in osm-3 (other PGPs not being discussed), but by RT-qPCR these three transporters exhibit a pretty mild upregulation in osm-3 and their expression is not reduced in nhr-8 osm-3 double mutants (Figure 5A). Can the authors provide an explanation for these discrepancies?

Specifically, the authors need to provide additional data to support the hypothesis that PGP upregulation is the basis of Tm resistance in osm-3 mutants. The authors can test whether genetic perturbations of different PGPs affect Tm sensitivity in osm-3 (dh441). Alternatively, they can construct verapamil resistant PGP mutants to demonstrate the specificity of the drug. If redundancy turns out to be a problem, the authors can try overexpressing pgps to suppress verapamil sensitivity. Even though redundancy could be a potential issues, silencing of a single PGP has been shown to be sufficient to sensitize animals to ivermectin (Menez et al., 2019). Additionally, Kurz et al. 2007 observed clear effects for pgp-5 mutants. Finally, verapamil resistant PGP variants have been reported in murine homologs (Loo and Clarke, 2001).

2) NHR-8 involvement in PFP upregulation and Tm resistance: Experiments with nhr-8 mutants should be better controlled to verify specific involvement in the tested phenotypes. Disruption of nhr-8, which the authors show is impairing pgp gene expression, is also important for *C. elegans* cholesterol metabolism, and could affect many additional genes; with regard to the impaired development of nhr-8 mutants (Figure 5B), alone or in combination with osm-3. It was shown that nhr-8 disruption increased dauer frequency (especially in combination with mild stress), which is likely to have contributed to the failure to develop on Tm (Magner et al. 2013). Therefore, it's important that the authors test whether ectopic expression of pgps rescues the Tm sensitivity of the nhr-8 and nhr-8 osm-3 double mutants.

3) UPR response in osm-3 (dh441): The authors emphasize that the lack of UPR induction they observe upon Tm treatment is due to a drug detoxification mechanism. However, they don't demonstrate this by confirming that the UPR can still be induced in these animals when activated by non-drug means eg. genetic activation of the UPR by expression of misfolded proteins, or by expression of spliced XBP-1. This is needed to confirm that drug detoxification is in operation here, rather than another mechanism, such as a reduction in the rate of translation, or a mechanism that increases ER homeostasis independent of the UPR. In addition, is this effect specific to Tm among the drugs that induce UPR activation, or is it a general effect? For example, does treatment with thapsigargin or another UPR-activating drug induce the UPR in osm-3 animals?

Similarly, Lines 243-245: "Instead of an activated stress signaling status that might explain the TM resistance, osm-3 Dyf mutants show no ER UPR induction upon TM treatment."

The implications of this point might be worth additional discussion / speculation. Do the data suggest that UPR overactivation could be a mechanism of TM toxicity, and that osm-3 mutants prevent that overactivation? Or do the authors favor the view that osm-3 mutants detoxify TM in such a way that ER UPR never needs to get activated in the first place?

[Editors' note: further revisions were suggested prior to acceptance, as described below.]

Thank you for resubmitting your work entitled "NHR-8 regulated P-glycoproteins uncouple xenobiotic stress resistance from longevity in chemosensory *C. elegans* mutants" for further consideration by *eLife*. Your revised article has been evaluated by Vivek Malhotra (Senior Editor) and a Reviewing Editor.

The manuscript has been improved but some issues remain that need to be addressed before acceptance, as outlined below:

1. The demonstration that pgp-8 KD was sufficient to increase hsp-4 expression in osm-3 mutants (Figure 5E, which is not cited in the main text, and not referred in the abstract) provides some support for the proposition that NHR-8 induction of pgp genes is responsible for TM resistance in osm-3 mutants. However, this is shown with higher TM concentration, sufficient to induce hsp-4 up-regulation even in control-treated osm-3 mutants, so that sensitization was a matter of scale rather than from nothing to something, which would have been more conclusive. Stronger support would have been provided by experiments examining the ability of pgp-8 RNAi to abolish developmental resistance to TM in osm-3 mutants (and to Ivermectin and paraquat to test the generality of detoxification mechanisms). Furthermore, additional experiments examining the effects of pgp-8 RNAi on worms over-expressing NHR-8 are important to establish that OSM-3, NHR-8 and PGP-8 function as one pathway.

2. Cholesterol experiments are rather indirect and therefore need to be interpreted carefully. The authors need to provide greater clarity in the interpretation of these data and what can or can't be concluded from them.

---

## [Author Response]

Essential revisions:1) PGP involvement in Tm resistance: The importance of PGPs rests upon the observation that treatment with verapamil reduces animals' resistance to Tm. However, the specificity of verapamil is not demonstrated in the paper. This drug has been shown to block calcium channels, which may affect diverse processes. More generally, the degree of PGP overexpression in osm-3 is pretty mild: is it reasonable to conclude that degree of overexpression is sufficient to produce the observed Tm resistance? Another concern is the inconsistency between the RNA-seq and RT-qPCR data for pgp expression. For example, by RNA-seq, pgp-2, pgp-9 and pgp-14 are reported to have significant difference in expression (Figure 4A and 4B) in osm-3 (other PGPs not being discussed), but by RT-qPCR these three transporters exhibit a pretty mild upregulation in osm-3 and their expression is not reduced in nhr-8 osm-3 double mutants (Figure 5A). Can the authors provide an explanation for these discrepancies?

Indeed, verapamil also blocks calcium channels, but the affinity is considerably lower compared to the affinity to PGPs. Matsuoka et al. reported an IC50 of verapamil of 0.6µM in cardiac cells (PMID: 2050292) and the reported calcium channel inhibition in worms is likewise reported with IC50 values in the μM range (PMID: 32208362). We used 1 nM verapamil, an at least 600fold lower concentration, with strong effects on TM resistance. We therefore concluded that the observed verapamil effects are due to PGP inhibition.

Our qPCR analysis showed a 5-10-fold elevation of 5 PGP mRNAs in osm-3 mutants (Figure 5A). Based on the new observation that single knockdown of the elevated pgp-8 was sufficient to reverse the TM-induced ER stress response we are now confident that these PGPs are responsible for the TM resistance.

Indeed, the RNA sequencing experiment failed to detect the elevated PGP levels that we later found by qPCR that are now functionally validated. Therefore, we can now be certain that the PGPs are induced and necessary for TM resistance. We have re-arranged the figures to reflect this.

Specifically, the authors need to provide additional data to support the hypothesis that PGP upregulation is the basis of Tm resistance in osm-3 mutants. The authors can test whether genetic perturbations of different PGPs affect Tm sensitivity in osm-3 (dh441). Alternatively, they can construct verapamil resistant PGP mutants to demonstrate the specificity of the drug. If redundancy turns out to be a problem, the authors can try overexpressing pgps to suppress verapamil sensitivity. Even though redundancy could be a potential issues, silencing of a single PGP has been shown to be sufficient to sensitize animals to ivermectin (Menez et al., 2019). Additionally, Kurz et al. 2007 observed clear effects for pgp-5 mutants. Finally, verapamil resistant PGP variants have been reported in murine homologs (Loo and Clarke, 2001).

To address the importance of the PGPs, we have now added genetic evidence that supports the initial pharmacological data with verapamil. We used RNAi against the upregulated pgp-8 and measured UPR signaling using the sensitive hsp-4::GFP reporter strain. As previously shown in the manuscript, this reporter is strongly induced by TM in the wildtype, but was not elevated at all in the osm-3 mutants. Our new data show that knocking down pgp-8 reversed the osm-3 resistance to TM thereby leading to hsp-4 induction in osm-3 mutants (Figure 5E). We conclude that the PGP upregulation is the basis for the TM resistance. Indeed, we were surprised that a single knockdown achieved a significant hsp-4 induction in osm-3 mutants.

This is potentially due to differences in tissue expression of the various PGPs.

2) NHR-8 involvement in PFP upregulation and Tm resistance: Experiments with nhr-8 mutants should be better controlled to verify specific involvement in the tested phenotypes. Disruption of nhr-8, which the authors show is impairing pgp gene expression, is also important for *C. elegans* cholesterol metabolism, and could affect many additional genes; with regard to the impaired development of nhr-8 mutants (Figure 5B), alone or in combination with osm-3. It was shown that nhr-8 disruption increased dauer frequency (especially in combination with mild stress), which is likely to have contributed to the failure to develop on Tm (Magner et al. 2013). Therefore, it's important that the authors test whether ectopic expression of pgps rescues the Tm sensitivity of the nhr-8 and nhr-8 osm-3 double mutants.

We thank the reviewers for this valuable comment regarding the specific role of nhr-8 in the control of the PGPs. Indeed NHR-8 is involved in cholesterol metabolism and its putative ligand is likely derived from cholesterol (Magner et al. PMID: 23931753). We therefore asked if NHR8 and downstream PGP expression as well as TM resistance might be affected by cholesterol. Indeed, pgp-8 was significantly upregulated in wildtype animals on high cholesterol. Further, we checked if PGP-dependent TM resistance might be affected by cholesterol. In both wildtype animals and osm-3 mutants, cholesterol was protective against TM toxicity. This protection was fully nhr-8 dependent. Together this suggests that upstream cholesterol or related metabolites act via NHR-8 and PGPs to modulate TM resistance.

We have spent considerable time and effort to generate PGP transgenic worms for overexpression in the nhr-8 and nhr-8; osm-3 double mutant. This would test if pgp expression is sufficient for TM resistance even in the absence of NHR-8. Unfortunately, despite our longstanding experience and multiple parallel approaches we did not succeed to generate pgp overexpressor *C. elegans* strains, potentially due to toxicity.

Nonetheless, our new data address the reviewer’s question if TM resistance in osm-3 mutants was due to NHR-8 regulation of PGPs. Expanding to upstream modulation by cholesterol or its derived signals, we avoid disruption of nhr-8 that might have pleiotropic effects and we find elevated pgp-8 expression. We demonstrate nhr-8 dependent TM resistance that requires PGPs. Together, these data clearly support the idea that NHR-8 controls pgp expression to affect TM resistance.

While nhr-8 is involved in dauer formation, this did not interfere with the interpretation of our results. First, in the TM resistance assays, nhr-8 mutants die immediately after hatching from the protective eggs, as L1 larvae, clearly before the dauer decision. Second, nhr-8 overexpression is sufficient to protect from TM toxicity, positioning NHR-8, independently from assays that might be affected by dauer formation, as an upstream regulator of xenobiotic stress resistance.

3) UPR response in osm-3 (dh441): The authors emphasize that the lack of UPR induction they observe upon Tm treatment is due to a drug detoxification mechanism. However, they don't demonstrate this by confirming that the UPR can still be induced in these animals when activated by non-drug means eg. genetic activation of the UPR by expression of misfolded proteins, or by expression of spliced XBP-1. This is needed to confirm that drug detoxification is in operation here, rather than another mechanism, such as a reduction in the rate of translation, or a mechanism that increases ER homeostasis independent of the UPR. In addition, is this effect specific to Tm among the drugs that induce UPR activation, or is it a general effect? For example, does treatment with thapsigargin or another UPR-activating drug induce the UPR in osm-3 animals?

We appreciate the reviewers’ points and agree that this is a key question: is the UPR still functional in osm-3 mutants in which we do not observe any TM-induced UPR signaling?

We addressed this question in two ways:

1) We induce ER stress genetically by knocking down the ERAD gene sel-1. This led to defects in clearance of misfolded proteins in the ER and triggered the UPR. Importantly, sel-1 RNAi induces hsp-4::GFP expression in osm-3 mutants.

2) We induced ER stress using a chemical approach exposing worms to the thiol reductase DTT. Just as the wildtype, osm-3 mutants mount a significant UPR response to DTT, as measured by increased xbp-1 splicing and hsp-4 mRNA levels. Consistent with this, motility was severely reduced in both wildtype and osm-3 animals after DTT treatment.

Similarly, Lines 243-245: "Instead of an activated stress signaling status that might explain the TM resistance, osm-3 Dyf mutants show no ER UPR induction upon TM treatment."The implications of this point might be worth additional discussion / speculation. Do the data suggest that UPR overactivation could be a mechanism of TM toxicity, and that osm-3 mutants prevent that overactivation? Or do the authors favor the view that osm-3 mutants detoxify TM in such a way that ER UPR never needs to get activated in the first place?Our data favor the second view: Due to detoxification, TM does not lead to UPR-ER activation. Our experiments did not address, and we cannot rule out that TM toxicity in the WT is in fact mediated by overactivated UPR. However, given the disruption of N-glycosylation by TM, the toxicity, in our view, is likely caused by a massive loss of function in the secretory pathway. However, the two views are somewhat connected: regardless of TM toxicity through run-away ER UPR or massive disruption of protein processing, osm-3 mutants are protected and our data show that this is mediated by PGPs. Additionally, it is unlikely that osm-3 mutations counter TM toxicity by dampening the UPR. Then, TM would still interfere with protein processing in the ER, leading to damage.

[Editors' note: further revisions were suggested prior to acceptance, as described below.]

The manuscript has been improved but some issues remain that need to be addressed before acceptance, as outlined below:1. The demonstration that pgp-8 KD was sufficient to increase hsp-4 expression in osm-3 mutants (Figure 5E, which is not cited in the main text, and not referred in the abstract) provides some support for the proposition that NHR-8 induction of pgp genes is responsible for TM resistance in osm-3 mutants. However, this is shown with higher TM concentration, sufficient to induce hsp-4 up-regulation even in control-treated osm-3 mutants, so that sensitization was a matter of scale rather than from nothing to something, which would have been more conclusive. Stronger support would have been provided by experiments examining the ability of pgp-8 RNAi to abolish developmental resistance to TM in osm-3 mutants (and to Ivermectin and paraquat to test the generality of detoxification mechanisms). Furthermore, additional experiments examining the effects of pgp-8 RNAi on worms over-expressing NHR-8 are important to establish that OSM-3, NHR-8 and PGP-8 function as one pathway.

We thank the reviewers for supporting our manuscript for publication. We have edited the final text to reflect the fact that the pgps likely act in concert and that individual knockdowns are expected to show only partial effects. We think that the effect of verapamil a the 1nM concentration is specific to PGPs and we have discussed the effects of higher concentrations in our reviewer response letter in the submission from September 2020. Therefore, we think that this is a direct pharmacological effect.

2. Cholesterol experiments are rather indirect and therefore need to be interpreted carefully. The authors need to provide greater clarity in the interpretation of these data and what can or can't be concluded from them.

We have carefully gone through the results and Discussion sections again and have made sure to limit our conclusions.